# Improved Automatic Deep Model for Automatic Detection of Movement Intention from EEG Signals

**DOI:** 10.3390/biomimetics10080506

**Published:** 2025-08-04

**Authors:** Lida Zare Lahijan, Saeed Meshgini, Reza Afrouzian, Sebelan Danishvar

**Affiliations:** 1Biomedical Engineering Department, Faculty of Electrical and Computer Engineering, University of Tabriz, Tabriz 51666-16471, Iran; 2Miyaneh Faculty of Engineering, University of Tabriz, Miyaneh 51666-16471, Iran; 3College of Engineering, Design and Physical Sciences, Brunel University London, Uxbridge UB8 3PH, UK

**Keywords:** BCI, CNN, graph theory, EEG, movement intention, finger tapping

## Abstract

Automated movement intention is crucial for brain–computer interface (BCI) applications. The automatic identification of movement intention can assist patients with movement problems in regaining their mobility. This study introduces a novel approach for the automatic identification of movement intention through finger tapping. This work has compiled a database of EEG signals derived from left finger taps, right finger taps, and a resting condition. Following the requisite pre-processing, the captured signals are input into the proposed model, which is constructed based on graph theory and deep convolutional networks. In this study, we introduce a novel architecture based on six deep convolutional graph layers, specifically designed to effectively capture and extract essential features from EEG signals. The proposed model demonstrates a remarkable performance, achieving an accuracy of 98% in a binary classification task when distinguishing between left and right finger tapping. Furthermore, in a more complex three-class classification scenario, which includes left finger tapping, right finger tapping, and an additional class, the model attains an accuracy of 92%. These results highlight the effectiveness of the architecture in decoding motor-related brain activity from EEG data. Furthermore, relative to recent studies, the suggested model exhibits significant resilience in noisy situations, making it suitable for online BCI applications.

## 1. Introduction

The brain–computer interface (BCI) allows direct interaction between the human brain and external equipment. Electroencephalography (EEG) data utilized in motor imagery brain–computer interfaces (BCIs) enable users to accomplish a variety of tasks without requiring physical movement [1]. EEG signals are used in many ways to detect emotions, detect sleep stages, detect driver fatigue, detect epilepsy, and detect depression [2]. This approach’s impact on the rehabilitation of persons with disabilities has raised it to a prominent interdisciplinary issue in recent years. Motor imagining (MI)-EEG analyzes and interprets signals from imagined tasks to control peripherals, wheelchairs, and prostheses [1].

The foundation for the majority of BCIs is established by evoked activity paradigms, including visually evoked steady-state potentials (SSVEPs) [3,4], event-related potentials (ERPs) [5], and motor-related paradigms like motor imagery [6]. Visual and attentional processes are necessary for the reliable elicitation of a quantifiable response in SSVEP and ERP. Conversely, movement neural correlates enable the voluntary generation of movement intents without the need for external inputs, thereby facilitating the intuitive control of BCIs [7,8]. Power changes across numerous EEG frequency bands are employed to assess movement intent. This method disregards the movement-related information present in the broader EEG spectrum and temporal domain due to the non-stationarity of the EEG signal. Neural movement correlates, such as motor-related cortical potential (MRCP) and event-related synchronization (ERD/S), are frequently employed to assess voluntary movement intention, execution, and visualization using EEG [9]. ERD and ERS are frequently employed to assess movement intention and imaging, resulting in a decrease in μ [10] and β [11] power and an increase in power. In order to ascertain tasks associated with movement, numerous features are extracted from the EEG spectral domain. The most prevalent method of evaluating ERD is through the analysis of power spectral density (PSD) and time frequency [12,13]. A progressive negative cortical potential, or MRCP, is detected at low frequencies and manifests approximately two seconds prior to voluntary movement. Identification is challenging due to the fact that the amplitude of MRCP is small (8–10 μV) in comparison to spontaneous EEG activity (100 μV) [14]. The average of numerous voluntary movement EEG samples is a widely used method for determining MRCP. In the following, recent studies of computational techniques for evaluating and tracking movement intention that are automatically developed based on EEG data are reviewed.

Haw et al. [15] used a single-channel EEG signal to automatically identify the movement intentions of five healthy individuals. The movement intention was categorized using the BP component. Two-stage classification was based on error thresholds and correlation. The technique has an accuracy rate of 70%. One of the study’s weaknesses was the difference in the proposed method’s performance among people. The use of a single-channel EEG signal in their study proved to be favorable. Yom et al. [16] employed a sample of five healthy adults to determine movement intention automatically. EEG waves were employed in nine channels during the experiment. To capture the signal, a finger-tapping action was also used. The researchers employed the MRP component to characterize movement intentions. The part was pre-processed using a low-pass filter set at 10 Hz. Categorization was accomplished using the K-nearest neighbor (KNN) and support vector machine (SVM). Bai et al. [17] conducted an experiment with 12 participants to assess the automated identification of movement intention. They recorded using 122 channels of EEG signals. Finger tapping provided the framework for the movement performed in their experiment. The researchers employed the MRP and ERD components to characterize movement intentions. The part was pre-processed using a Butterworth low-pass filter of the third order. The classification accuracy in the two stages using artificial neural networks (ANNs) was 75%. One of the method’s disadvantages is that it employs 122 EEG signal channels, which may be unpleasant for patients and increase power consumption in prosthetic devices. Kato et al. [18] used a single-channel EEG signal to automatically identify the purpose of movement in seven healthy persons. Their experiment revealed tapping-based movement. The contingent negative variation (CNV) component was used to classify movement intentions. For classification, the SVM was utilized. Boye et al. [19] employed a single volunteer to automatically determine movement intention. The EEG signal was obtained by tapping the finger. The researchers employed the MRP component to characterize movement intentions. In the pre-processing step, a low-pass filter and principal component analysis (PCA) were applied. The classification challenge was performed using the KNN and SVM algorithms. They found a 96% classification sensitivity for the two stages. The testing on a single subject was a limitation of the study. Lew et al. [20] utilized eight healthy participants and two stroke survivors to automatically measure movement intention. For signal recording, sixty-four EEG channels were employed. They employed arm motion as the foundation for movement in their experiment. In the pre-processing step, an IIR filter with a cutoff frequency of 0.1 was used. The KNN was used to categorize data. Their technique of distinguishing movement intention was shown to be effective 76% of the time. According to studies, the proposed algorithm was effective 82% of the time for healthy participants and 64% of the time for sick individuals. In the study by Niazi et al. [21], 16 healthy volunteers’ movement intentions were automatically recognized. The data were collected utilizing 10 channels of EEG signals. Leg movement served as the foundation for the movement type in their study. The researchers employed the BP and MRCP components to classify movement intentions. For categorization, the Neyman–Pearson Lemma (NPL) was utilized. Niazi et al. [22] conducted studies on twenty healthy people and five stroke sufferers to automatically assess movement intention. The study utilized ten channels of EEG signals for re-recording and focused on limb movements. The researchers employed the MRP component to characterize movement intentions. During the data-processing step, a band-pass filter was applied in the frequency range of 0.05 to 10 Hz. Ahmadian et al. [23] conducted experiments with three healthy participants. The study acquired data for the automated identification of movement intention utilizing 128 channels of EEG signals. The signal was captured using a tapping motion with the fingertips. The researchers employed the BP component to characterize movement intentions. The pre-processed component used an ideal filter with a frequency range of 0.5 to 70 Hz. Furthermore, the dimensionality of the feature vector was reduced using the independent component analysis (ICA) technique. It took around 51 s for the algorithm to discriminate between the blind sources. The study’s limitations were the huge number of channels employed in the EEG signal and the small number of samples collected. In order to automatically recognize movement intention, Jochumsen et al. [24] conducted an experiment with 12 healthy volunteers. Moreover, the signal was recorded using 10 channels of EEG signals. Additionally, the way they moved was determined by how their legs moved during the trial. In the study, 0.5 to 10 Hz was the perfect filter they employed for the pre-processed portion. The feature-vector dimension was decreased by the researchers using the constraint–satisfaction–problem (CSP) technique. For the classification, SVM was also employed. Their approach was claimed to have an 80% overall effectiveness rate in differentiating movement intention. Xu et al. [25] included nine healthy participants. To capture the signal, they used nine EEG wave channels. They used the MRCP component in the experiment, and their movement style was also determined by foot movement. To pre-process the data, they used a band-pass filter with frequency ranges of 0.5 to 3 Hz. They claimed 75% classification accuracy for the first two phases using KNN. Jiang et al. [26] studied nine healthy people. They used nine EEG channels to record the signal for automatically recognizing movement intention. Furthermore, they used the MRCP component to classify movement intention and determined the movement type based on leg movement during the trial. They increased the SNR by using LSF. Their two-stage classification was estimated to be 76% accurate. Wairagkar et al. [27] recruited nine healthy participants, six men and eight women, aged 22 to 30. The autocorrelation function was used in this analysis. The researchers used the ERD component to categorize movement intentions. The categorization process also made use of the KNN. They claimed that their two-stage classification was 78% sensitive. Shahini et al. [28] presented a new method for automatically detecting movement intention using EEG signals. They used basic convolutional neural networks (CNNs) for feature selection/extraction and classification, achieving significant accuracy in two and three different classes of finger strokes. Their network architecture included 10 convolutional layers and two fully connected layers. Jochumsen et al. [29] used EEG and electromyography (EMG) signals to automatically detect movement intention in Parkinson’s disease patients. These researchers used engineering methods for feature selection/extraction and were able to collect and classify a database in three distinct scenarios. The three-class mode has the highest reported classification accuracy of around 89%. Lutes et al. [30] used EEG signals to automatically detect movement intentions. In this study, they combined convolutional networks and achieved an accuracy of 98.50 for negative skew. When compared to other networks like EEGNet and GraphNet, their model outperformed them all. Choi et al. [31] used EEG signals to classify movement intentions. They selected and extracted features from an offline dataset using a proposed pipeline. They also used the SVM classifier for classification. The final classification accuracy was reported to be 86%. Dong et al. [32] proposed automatically detecting movement intention using EEG signals. These researchers used transfer learning to identify movement intention based on the affected arm’s bidirectional movement. They collected a database of 12 healthy individuals who were able to recognize movement intention using machine learning techniques on virtual reality (VR) induction. The average accuracy reported in this study was 85%.

Analysis of prior studies indicates that the majority necessitate elevated EEG channels for optimal efficacy. This issue can enhance the computational efficiency of the algorithm and render its application in artificial prostheses unfeasible. Furthermore, the processes of feature selection, extraction, and classification in the majority of studies are conducted manually and through engineering techniques. This necessitates prior understanding of the issue and is inappropriate for BCI applications. Moreover, research utilizing deep learning is not without its limitations. A fundamental limitation of these studies is the absence of a substantial database for network training, as deep learning networks require extensive data. Furthermore, owing to the low SNR ratio of EEG signals, the robustness of the deep models discussed in studies concerning movement intention in noisy environments has not been assessed. This research aims to address the challenges associated with recent studies and offer a dependable method for the automatic classification of movement intention. In this study, a comprehensive database of EEG signals was collected under three distinct conditions in two scenarios: resting state, right finger movement, and left finger movement. Upon completion of the pre-processing phase, the recorded data will proceed to the automated feature selection and extraction phase, utilizing a combination of graph theory and deep convolutional networks to classify various categories within the specified scenarios.

A.Preparation of a database of EEG signals during movement intention testing in two separate scenariosB.Automatic presentation of an intelligent system in the automatic classification of movement intention based on the combination of graph theory and deep convolutional networksC.Presenting a new model with high speed and accuracy for classifying left finger stroke, right finger stroke, and the resting stateD.Achieving the highest classification accuracy for the two-class mode compared to recent researchE.Ability to apply the algorithm in noisy environments in order to use the proposed model in online applications

The article’s remaining sections are arranged as follows: The model used in this study is mathematically analyzed in the Section 2. The proposed model used in this study is thoroughly described in the Section 3, which also includes a suggested architecture, data-recording techniques, and other relevant information. The simulation results are shown in the Section 4 along with a comparison with current research findings. The conclusion is covered in the Section 5.

## 2. Materials and Procedures

In this section, the mathematical basis of the algorithms used in this research, which include generative adversarial networks (GANs) and graph neural networks (GNNs), is fully examined.

### 2.1. Generative Adversarial Networks (GANs)

In 2014, Ian J. Goodfellow and his associates proposed the GAN. In machine learning, GANs perform unsupervised learning tasks. These networks are made up of two models that recognize and incorporate the patterns in the input data on their own. The generator and discriminator are the names given to these two models. The discriminator and the generator compete to identify, record, and reproduce changes made to the dataset. New samples that are statistically representative of the original dataset can be generated by GANs [33].

A neural network functions as the generator, creating artificial data for the discriminator’s training. The generator gains the capacity to produce adequate data. For the discriminator, the generated instances are considered negative training examples. The generator creates a sample after receiving a fixed-length random noise vector as input. The generator’s main goal is to fool the discriminator into thinking that its output is “genuine”. The component of the GAN responsible for training the generator includes (a) a noisy input vector; (b) a generative network that converts the random input into a data sample; (c) a discriminative network, which categorizes the produced data; and (d) generative losses, which penalize the generative network for perceiving the differentiator as foolish.

The backpropagation algorithm adjusts each weight appropriately by assessing the impact of the weight on the output. This method is employed to acquire gradients, which can facilitate modifications to the productive weights.

A discriminator is a neural network that differentiates authentic data from synthetic data produced by the generator. The training data for the differentiator is sourced from two distinct origins: (a) The discriminator utilizes authentic data samples, including medical images, medical signals, human subjects, and currency notes, as positive examples in the training process. (b) During the training process, counterfeit samples produced by the generator are utilized as negative samples.

Throughout the training process, the discriminator is linked to two loss functions. In the training of the discriminant network, productive losses are disregarded, and solely discriminant losses are utilized. The discriminator, throughout the training process, classifies the authentic data and the fabricated data provided by the generator. The discriminant loss penalizes the misclassification of a genuine data sample as a counterfeit sample or vice versa. The discriminator adjusts its weights by backpropagating the losses through its network.

In GANs, the following equation is minimized in the training stage:(1)$$\log(1-D(G_{\left(Z\right)}))\;\begin{array}{ll} \frac{\min\max}{GD}V(G,D)=E_{x} & -P_{data}[\log D(x)]\\ & +\;E_{pz(z)}[\log(1-D(G(Z))] \end{array}$$

To properly distinguish between real and fake data, the discriminator (D) in the above equation needs to be set up. The aforementioned equation requires iterative algorithms because it cannot be solved in a closed form. In order to address the problem of overfitting, the generator function (G) is also optimized once for every k function D optimization [33].

### 2.2. Graph Convolutional Network

A GNN is designed to operate diagonally on graphs, which are data structures composed of nodes (also known as vertices) and the edges that connect them [34]. GNNs have radically revolutionized how we use and assess graph-organized data.

GNNs are typically used to learn an embedding of the graph structure, in which the GNN records the features of the nodes (i.e., what they contain) as well as the topology of the graph. These representations may then be used to perform various tasks such as classifying whole graphs, predicting the existence of an edge between two nodes, and determining a node’s label. The following section discusses some of the themes linked to GNNs [34].

Vertices and edges: A graph comprises a collection of points (vertices) interconnected by lines (edges). Vertices denote entities, objects, or concepts, whereas edges signify relationships or connections among them. Directed versus Undirected: In a directed graph, the edges possess a direction that signifies the flow of the relationship. Weighted Graph: In these graphs, edges possess associated weights. Graph representation encodes the structure and attributes of a graph for neural network processing. Graphs incorporate node data along with the interconnections between data points. A graphical representation is required to illustrate the connections among nodes. Presented below are several prevalent graph representations utilized in deep learning. Adjacency Matrix: This matrix enumerates all vertices that are connected to a specific vertex (all nodes linked to a node). Incidence Matrix: An *N* × *M* matrix where *N* represents the number of nodes and *M* denotes the edges of the graph. It is utilized to represent the graph as a matrix. The value is 1 if the node possesses a specific edge and 0 if it does not. Degree Matrix: A diagonal matrix that enumerates the edges associated with each node [35].

An adjacency matrix is employed to connect each vertex in the graph. Furthermore, the degree matrix can be derived from the adjacency matrix. The diagonal elements of this diagonal matrix correspond to the sum of the edges connected to the respective vertex. The degree matrix is denoted as $D\in R^{N\times N}$ and the graph matrix as $W\in R^{N\times N}$, with the *i*-th diagonal element of the degree matrix defined as follows:(2)$$D_{\mathrm{ii}}=\sum_{i}W_{ij}$$

An alternative definition of the Laplacian matrix is as follows:(3)$$L=D-W\in R^{N\times N}$$(4)$$L=U\Lambda U^{T}$$

The Laplacian matrix is known to be formed by subtracting the degree matrices from the adjacency matrix, as per the preceding relation. Graph basis functions are calculated using this matrix. In the Laplacian matrix, Singular Value Decomposition (SVD) may be used to produce graph basis functions. Additionally, the matrix of singular values and the matrix of eigenvectors in the form of Relation (5) may be used to define the Laplacian matrix. The columns of the eigenvector matrix match the eigenvectors of the Laplacian matrix, as stated in Equation (5). Based on these eigenvectors, it is also feasible to perform a Fourier transform. The diagonal eigenvalues, which include $\Lambda =diag([\lambda_{0,\ldots ,}\lambda_{N-1}])$ in the connection shown below, determine the Fourier bases:(5)$$U=[u_{0},\ldots ,u_{N-1}]\in R^{N\times N}$$

To enhance comprehension, Relations (7) and (8) define the Fourier transform and inverse Fourier transform of a signal, such that(6)$$\hat{q}=U^{T}q$$(7)$$q=UU^{T}q=U\hat{q}$$

Equation (7) states that $\hat{q}$ stands for the graph’s Fourier transform. Additionally, the feature vector for a signal like q with Fourier bases and the graph’s Fourier transform is feasible, according to Equation (8). Another method for calculating the graph convolution operator is to use the Fourier transform of each signal to perform a convolution of two signals in the graph domain. For ease of comprehension, the relationship between the convolution of two signals, *z* and *y*, and the operator ${*}_{g}$ is as follows:(8)$$z{*}_{g}=U((U^{T}z)⊙(U^{T}y))$$

In the connection above, a graph convolution operator combined with neural networks is described by the g() filter function. *z* is the version that g(L) filtered, based on the relationship mentioned above:(9)y=g(L)z

The following definition of graph convolution may be obtained by setting the Laplacian matrix and breaking it down into singular values and eigenvectors [34,35]:(10)$$\begin{array}{l} y=g(L)z=Ug(\Lambda )U^{T}z\\ =U(g(\Lambda ))⊙(U^{T}z)\\ =U(U^{T}(U_{g}(\Lambda )))⊙(U^{T}z)\\ =z{*}_{g}(Ug(\Lambda )) \end{array}$$

## 3. Proposed Model

This section delineates the suggested methodology of this work for detecting movement intention from EEG data. This section addresses the procedures for database recording, data pre-processing, network architecture design, optimization of architectural parameters, and the allocation of training and testing data. Figure 1 visually illustrates the proposed flowchart of the investigation.

### 3.1. Data Acquisition

This study involved the collection of an extensive database utilizing EEG signals for the automatic classification of movement intention. Sixteen undergraduate and graduate students (eight women and eight men), aged between 20 and 33 years with an average BDI of 22, were solicited to partake in the signal recording experiment for movement intention. The experiment was described to all participants, and informed consent was obtained from them. Furthermore, ethics permit number IR.TBZ.1399,5,4 was granted by the ethics committee of Tabriz University for the documentation of EEG data. An open BCI amplifier from an American company was utilized in this test to record the signal according to the 10–20 standard. The sampling frequency of this 21-channel amplifier was 1024 Hz for signal recording, with A1 and A2 channels utilized as references. This experiment aimed to classify three states: resting, right-hand finger tapping, and left-hand finger tapping. Consequently, two distinct scenarios were evaluated for classification. The initial scenario comprises classes for right-hand finger taps and left-hand finger taps. The second scenario encompasses right-hand finger strikes, left-hand finger strikes, and a resting state.

The signal recording had 40 repetitions, the length of each mode was 5 s, and for each mode, 5 × 1024 = 5120 sampling points with 40 repetitions were available. Among the participants in the experiment, 12 were right-handed and 4 were left-handed. According to studies [27,28], only 6 pairs of electrodes were considered for signal recording, which included F3-C3, Fz-Cz, F4-C4, C3-P3, Cz-Pz, and C4-P4, and the rest of the electrodes were not used for recording and processing. This work reduces the computational complexity for the classification operation to a significant extent. Thus, the dimensions of the data for each class of left finger tap, resting state, and right finger tap were equal to 40 (repetitions) × 5120 (sampling points) = 204,800 per participant. The devices used for testing are shown in Figure 2.

### 3.2. Pre-Processing of EEG Data

All processing in this study was performed offline. This sub-section contains the pre-processing pertinent to this study. Before the data gathered in this bulletin can proceed to the classification processing stage, it must undergo pre-processing. This study includes pre-processing techniques such as the application of a notch filter, a second-order Butterworth filter, data enhancement, and data normalization. Subsequently, each of these steps is elucidated individually:I.To eliminate the interference caused by the 50 Hz frequency of municipal electricity, a notch filter was applied to the EEG data collected from the F3-C3, Fz-Cz, F4-C4, C3-P3, Cz-Pz, and C4-P4 channel pairs.II.The recorded data underwent processing through a second-order Butterworth filter, targeting the frequency range of 0.05 to 60 Hz for the respective channels of the recorded signals.III.The recorded data are augmented through GANs to mitigate the occurrence of overfitting. Data augmentation in the GAN is performed by the generator and the discriminator, as previously stated. The subsequent section will provide a comprehensive description of the data augmentation process utilizing the GAN. The generator and discriminator in the GAN execute data augmentation, as previously mentioned. A uniformly distributed 100-dimensional vector is transformed into a 1 × 204,800-dimensional signal by the generating network. The generator produces a one-dimensional signal with vector dimensions of 100, characterized by a uniform distribution. The generating network consists of six convolutional layers, each with dimensions of 512, 1024, 2048, 40,996, 8192, and 204,800. Batch normalization and Relu activation are utilized in each layer. The repetitions and learning rate are established at 150 and 0.01, respectively. The discriminative network receives a one-dimensional vector as input and assesses its authenticity. This network consists of six dense layers. Employing adversarial generative networks, the data is enhanced from 204,800 dimensions to 250,000 dimensions.IV.During the final phase, data normalization is executed to optimize the training process within the range of 0 to 1.

### 3.3. Graph Design

Following the determination of the functional relationship between the EEG channels, a proximity matrix is created. This may be achieved by assessing the channel correlation and showing the findings as an EEG channel connection matrix. To eliminate the network adjacency matrix, the sparse approximation of the connectivity matrix is set to a threshold. The recommended model utilizes the constructed graph as input to select, extract, and classify information. Figure 3 shows an overview of the proposed architecture.

### 3.4. Customized Architecture

This subsection delineates the deep architecture developed for classifying movement intention into two and three distinct classes. Figure 4 illustrates the intricately designed architecture. This figure indicates that following a dropout layer, the data arrives at the initial layer of the convolutional graph, accompanied by a max pooling layer and a batch normalizer utilizing the Leaky-Relu activation function. To select or extract the automatic feature, these layers are reiterated five additional times. The data is subsequently input into a dropout layer. Subsequently, the data undergoes a flattening operation. The classes pertaining to the right finger stroke, resting state, and left finger stroke are evaluated using a Softmax activation function to classify into two and three distinct categories.

The proposed deep architecture features a node graph representing the quantity of EEG channels considered. In the proposed architecture, each vertex receives 1000 samples. The coefficients of *X*_1_*–X*_6_ are presented as Chebyshev polynomial coefficients for each layer in Table 1.

### 3.5. Series of Tests, Validation, and Training

This subsection delineates the methodology for partitioning the data into training and evaluation sets. Specifically, 70% of the data is allocated for network training, 20% for the validation set, and 10% for the test set. The design methodology employs a trial-and-error approach concerning variables and algorithms. Consequently, Table 2 presents the parameters, variables, and various optimization algorithms pertinent to the development of the proposed deep architecture.

## 4. Experimental Results

This section will delineate the results pertaining to the proposed model. This section comprises multiple subsections: the first presents the results of the proposed network optimization, the second details the simulation results, and the third offers a comparison with recent research. The simulation results of this study were conducted on the Python version 3.10 programming platform within the Google Colab Prime environment, utilizing 32 GB of RAM and a 60 GPU.

### 4.1. Enhancing Outcomes

This subsection will present the optimization results derived from the proposed deep architecture. Figure 5 illustrates the outcomes pertaining to the determination of the number of layers in the proposed deep network. This figure indicates that the selection of six convolutional graph layers has proven optimal for accurately classifying movement intention. Thus, augmenting the number of layers minimally impacts classification accuracy while significantly elevating the algorithm’s computational complexity. Figure 6 illustrates the outcomes of Chebyshev polynomial selection for the classification of automatic movement intentions. Thus, selecting *X* = 5 can accelerate the network’s convergence to the target value.

### 4.2. Results of the Simulation

This subsection will present the simulation results of the proposed model.

Figure 7 illustrates the accuracy and error of movement intention classification across 150 iterations of the network for two classes (left finger tap and right finger tap) and three classes (left finger tap, resting state, and right finger tap). Figure 7a indicates that the network’s accuracy has attained 98% after 150 iterations for the classification of two classes and has stabilized. Furthermore, as indicated by the same figure, the accuracy for the three-class mode stabilizes at approximately 92% following 120 iterations. Figure 7b additionally illustrates the classification error for both the two-class and three-class modes. According to the same figure, as the number of repetitions for two-class and three-class classification increases, the network error attains its minimum value. Table 3 analyzes the outcomes pertaining to the assessment criteria for the binary and ternary classifications. This study employs evaluation criteria comprising accuracy, precision, sensitivity, specificity, and the kappa coefficient. In the two-class mode, all evaluation metrics exceed 97%, demonstrating the efficacy of the proposed deep network. Figure 8 illustrates the receiver operating characteristic (ROC) curve analysis for the classification of various classes. Consequently, it is established that for the classification, both scenarios fall within the range of 0.9 to 1, indicating the optimal performance in the automatic classification of movement intention across two and three distinct classes. Figure 9 illustrates instances of the left finger tap, right finger tap, and resting state in both the initial and final layers of the proposed deep network. It is evident that, in the two-class mode, nearly all samples are distinctly separated in the final layer of the network.

### 4.3. Comparison with Current Methodologies and Research

This subsection contrasts the proposed method with current methodologies and studies. Table 4 delineates the methodologies and accuracy of contemporary studies focused on classifying movement intention. The proposed model demonstrates enhanced accuracy compared to recent studies, attaining a classification accuracy of 92% for three classes, whereas studies [31,32] report accuracies of approximately 85% and 86%, respectively. It is essential to recognize that the databases employed in recent studies vary, making direct comparison unjust. The parameters for recording brain signals, participant numbers in each study, sampling frequencies, and additional factors differ among research studies. Based on the current evidence, to guarantee equitable comparative conditions, we utilized modern methodologies from the registered database and compared the results with our proposed model. This research utilizes pre-trained networks Inception [36], VGG [37], U-net [38], and basic CNN [28]. Figure 10 displays the results obtained for the classification of two categories. The proposed model has clearly converged to the target value more swiftly. Additionally, we implemented a supplementary comparative strategy employing manual feature selection/extraction and feature learning techniques. Thus, for the manual approach, the attributes of mean, variance, peak coefficient, power, kurtosis, and skewness were derived from the recorded EEG signals, and classifications were executed utilizing the KNN [39] SVM [40], multi-layer perceptron (MLP) [41], basic CNN [28], and the proposed method. The feature learning method entailed classifying the recorded signals without the use of feature selection or extraction, relying on the designated classifications. Table 5 delineates the comparative results of the manual technique and the feature learning approach. The proposed model employing a feature learning approach has exhibited an enhanced performance. Nonetheless, the proposed model demonstrates low accuracy in comparison to the manual method. The proposed model, which amalgamates graph theory and deep convolutional networks, autonomously and end-to-end learns salient features from recorded signals, differentiating between two and three classes. Manual methods, while simple, require prior understanding of the problem and may be impractical for online applications.

EEG signals exhibit a very low SNR. This problem can obstruct classification in online applications. Minimal motion and ambient noise can hinder the accurate discernment of movement intention. The employed classification algorithm must exhibit significant resilience to environmental noise. Thus, we have synthetically introduced Gaussian white noise with random dispersion into the signals recorded at various dB levels to assess the efficacy of our proposed deep model under noisy conditions. The results obtained are illustrated in Figure 11. The figure clearly illustrates that the proposed network exhibits a significantly lower slope of decreasing classification accuracy in response to increasing noise compared to the other networks analyzed. This illustrates the significance of integrating graph theory with deep convolutional networks.

As we know, the use of artificial intelligence [42,43,44,45], which is a subset of machine learning [46,47], has been very useful in various applications, including vibration [48], deep learning [49,50,51,52,53,54], fuzzy networks [55,56], financial markets [57,58,59,60,61], leadership [62], education [63,64,65], mathematics [66,67], architecture [68,69,70], history [71], civil engineering [72,73], optimization [74], economics [75,76,77], chemistry [78], law [79], aerospace [80], image processing [81], transportation [82,83,84,85], intelligent systems [86,87,88,89], supply chain [90], computers [91], business [92], etc.

Notwithstanding the exceptional performance, the proposed model exhibits certain deficiencies. The implementation of the proposed deep model in this research necessitates an expansion of the database dimensions. Furthermore, it is essential to employ classical overlay for data augmentation and evaluate its efficacy against GANs. Furthermore, to assess the proposed model in real-time settings, it is essential to utilize dry electrodes for signal acquisition to eliminate issues related to gel desiccation during recording.

## 5. Conclusions

This study introduces a novel method for the automatic detection of movement intention across two categories: left finger tap and right finger tap, as well as three categories: left finger tap, resting state, and right finger tap. To achieve this objective, EEG signals from 20 participants were collected during the movement intention test. Following essential pre-processing, feature selection/extraction and automatic classification were conducted utilizing a combination of graph theory and deep convolutional networks. The proposed network included six convolutional graph layers that could perform end-to-end classification operations. The classification outcomes of movement intention in this study are highly promising, even amidst environmental noise, and can rival the results of recent research. The suggested method is applicable in numerous domains within the field of BCI.

## Figures and Tables

**Figure 1 biomimetics-10-00506-f001:**
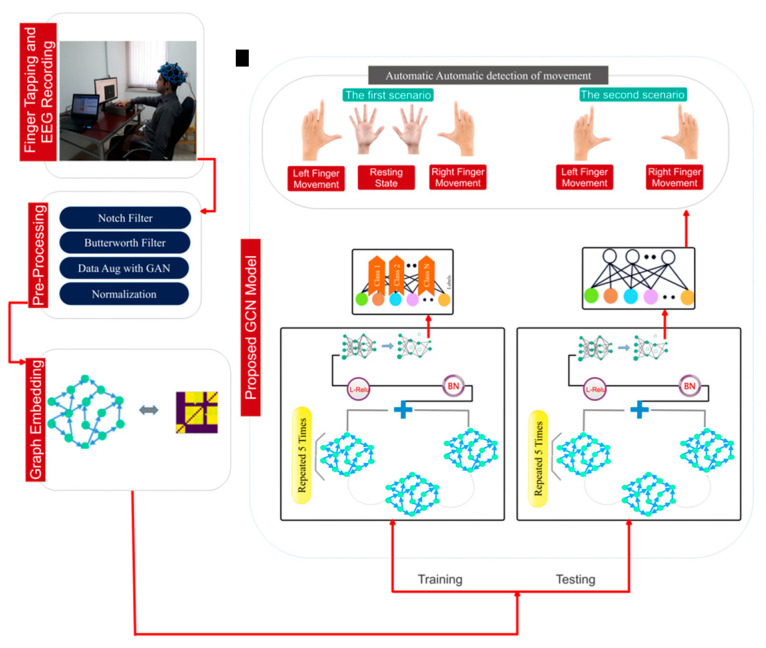
The main framework of this study used to automatically detect movement intention from EEG signals.

**Figure 2 biomimetics-10-00506-f002:**
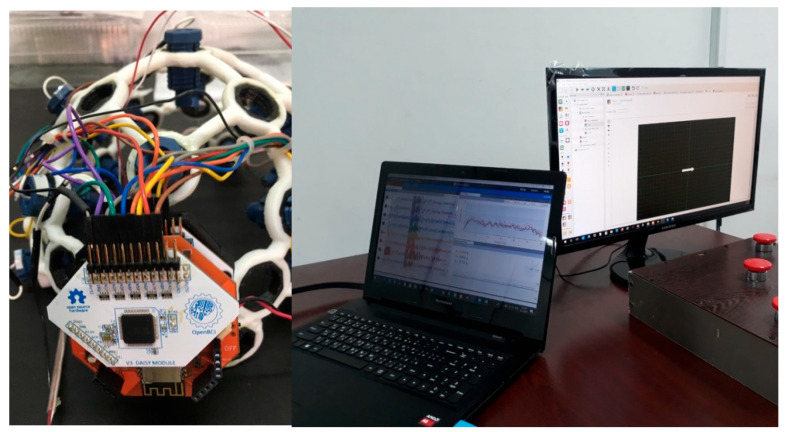
Example of a tapping device and recording of EEG signals.

**Figure 3 biomimetics-10-00506-f003:**
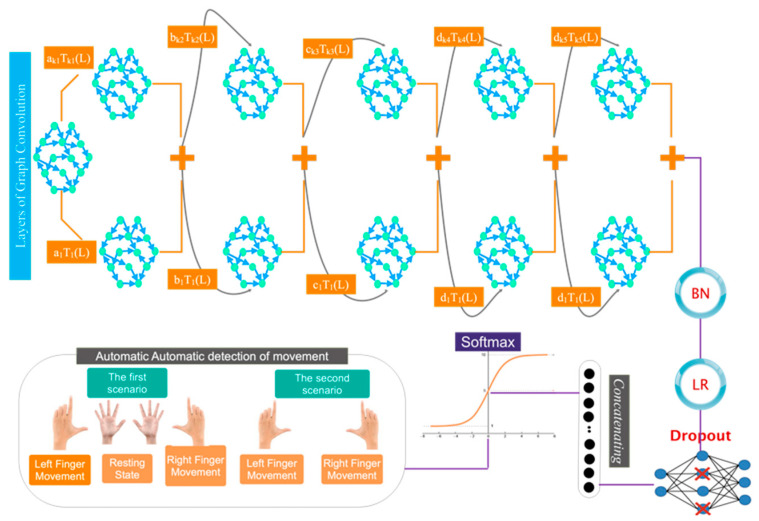
An overview of the architecture presented in this study.

**Figure 4 biomimetics-10-00506-f004:**
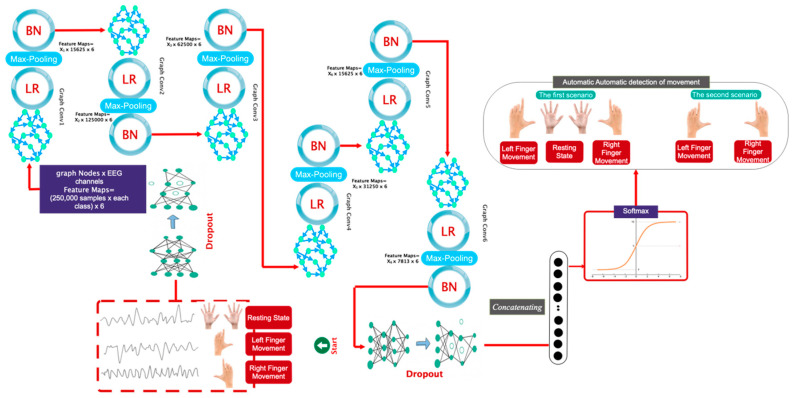
Detailed architecture of the proposed deep model along with dimensions of samples in each layer.

**Figure 5 biomimetics-10-00506-f005:**
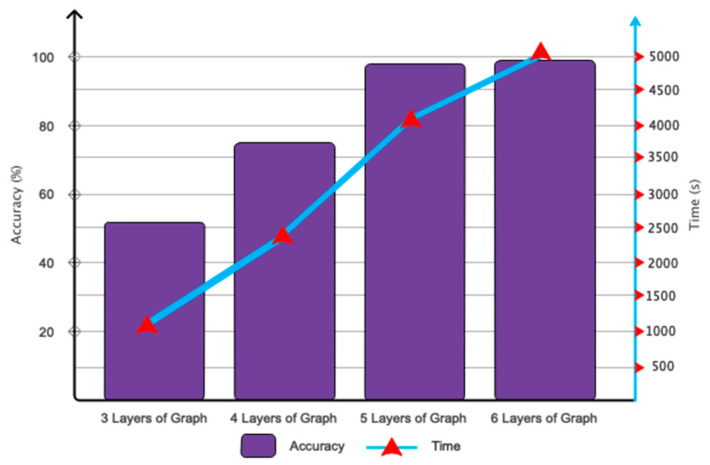
Accuracy and time related to movement intention classification considering different graphConv layers.

**Figure 6 biomimetics-10-00506-f006:**
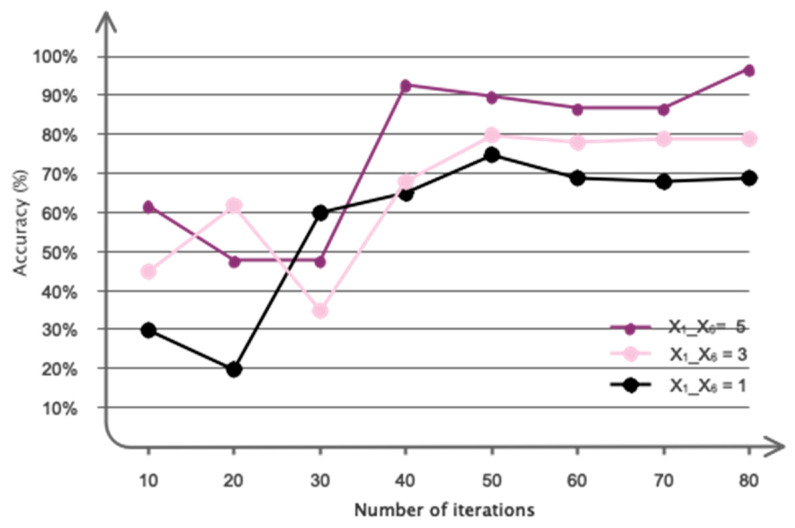
Classification accuracy of movement intention by considering different polynomial variables.

**Figure 7 biomimetics-10-00506-f007:**
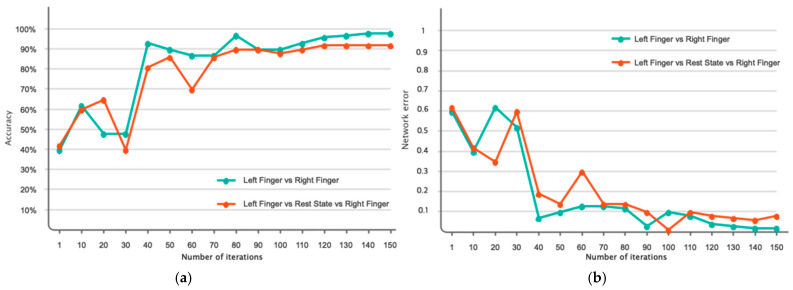
Accuracy and error of the proposed model in two-class and three-class scenarios for 150 network iterations. (**a**) Accuracy of network, (**b**) Loss function of network.

**Figure 8 biomimetics-10-00506-f008:**
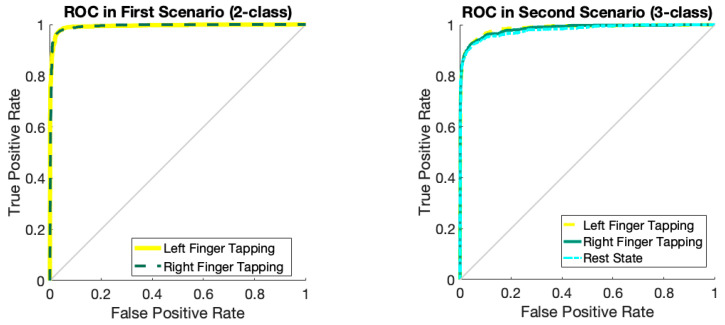
Analysis of ROC curves for different classes of movement intention.

**Figure 9 biomimetics-10-00506-f009:**
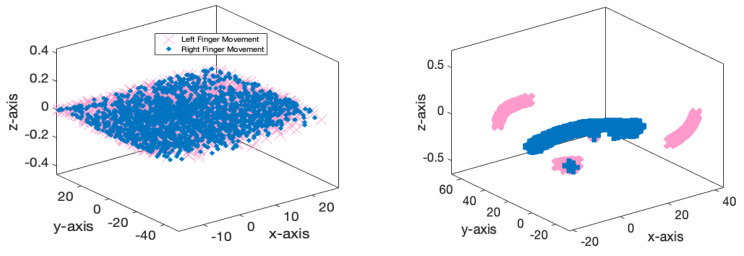
Recorded EEG samples related to left finger tap, right finger tap, and resting state in two different scenarios for the input and output of the proposed network.

**Figure 10 biomimetics-10-00506-f010:**
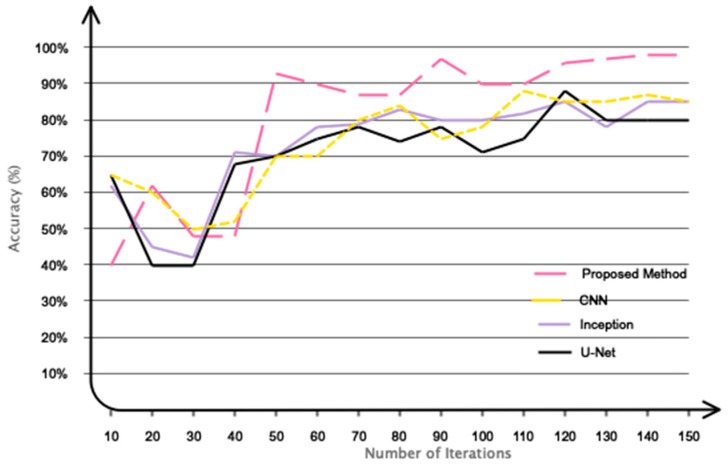
Evaluating the efficacy of the constructed network against pre-trained networks.

**Figure 11 biomimetics-10-00506-f011:**
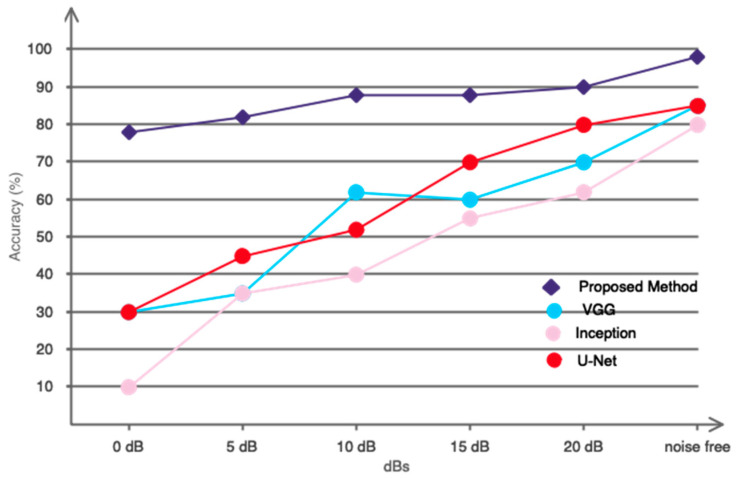
Evaluation of the intended deep network’s performance in noisy situations.

**Table 1 biomimetics-10-00506-t001:** Details about hyperparameters in the proposed deep architecture.

Layer	Shape of Weight Tensor	Shape of Bias	Number of Parameters
**Graph 1**	(x_1_, 250,000, 250,000)	250,000	62,500,000,000 × x_1_ + 250,000
**Graph 2**	(x_2_, 250,000, 125,000)	125,000	31,250,000,000 × x_2_ + 125,000
**Graph 3**	(x_3_, 125,000, 62,500)	62,500	7,812,500,000 × x_3_ + 62,500
**Graph 4**	(x_4_, 31,250, 15,625)	15,625	488,281,250 × x_4_ + 31,250
**Graph 5**	(x_5_, 15,625, 7813)	7813	122,078,125 × x_5_ + 15,625
**Graph 6**	(x_6_, 7813, 3907)	3907	30,525,391 × x_6_ + 7813
**Flattening Layer**	-	2	3907

**Table 2 biomimetics-10-00506-t002:** Selected variables in the proposed architecture.

Model	Parameters	Values	Optimal Value
**GAN**	Batch Size	4, 6, 8, 10, 12	10
	Optimizer	Adam, SGD, Adamax	Adam
	Conv layers	3, 4, 5, 6	6
	Learning Rate	0.1, 0.01, 0.001, 0.0001	0.01
	Number of GConv	2, 3, 4, 5, 6, 7	6
**ConvGraph**	Batch Size in DFCGN	8, 16, 32	32
	Batch Normalization	ReLU, Leaky-ReLU	Leaky-ReLU
	Learning Rate in DFCGN	0.1, 0.01, 0.001, 0.0001, 0.00001	0.0001
	Dropout Rate	0.1, 0.2, 0.3	0.1
	Weight of Optimizer	10 × 10^−3^, 10 × 10^−4^, 10 × 10^−5^, 10 × 10^−6^	

**Table 3 biomimetics-10-00506-t003:** Different evaluation indices used to automatically classify movement intention.

Measurement Index	Accuracy (%)	Sensitivity (%)	Precision (%)	Specificity (%)	Kappa Coefficient
**2-class**	98.1	97.4	97.4	97.8	0.88
**3-class**	92.2	91.7	89.4	91.4	0.81

**Table 4 biomimetics-10-00506-t004:** The suggested model is compared to other recent investigations.

Research	The Method Used	ACC (%)
**Jochumsen et al. [24]**	CSP + SVM	80
**Xu et al. [25]**	MRCP Component + KNN	75
**Jiang et al. [26]**	MRCP Component	76
**Wairagkar et al. [27]**	ERD Component + KNN	78
**Shahini et al. [28]**	CNN	89
**Jochumsen et al. [29]**	Hand Crafted Features + KNN	89
**Lutes et al. [30]**	CNN	98.50 (**two class**)
**Choi et al. [31]**	Hand Crafted Features + SVM	86
**Dong et al. [32]**	Transfer Learning	85
**Our Model**	**GAN + Graph Theory + CNN**	**98.2 (two class) 92 (three class)**

**Table 5 biomimetics-10-00506-t005:** Comparison of the proposed model with manual methods.

Method	Feature Learning (ACC)	Handcrafted Features (ACC)
**KNN**	76%	82%
**SVM**	80%	85%
**CNN**	84%	60%
**MLP**	75%	79%
**P-M**	92%	69%

## Data Availability

The original contributions presented in this study are included in the article material. Further inquiries can be directed to the corresponding authors. The image in Figure 1 was captured directly during the EEG data recording session of our study by the authors. It is an original image and not obtained from any external source or public database. The image is not available for public use and is restricted to this research purpose only.

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
