# Peer review of "Improved Automatic Deep Model for Automatic Detection of Movement Intention from EEG Signals"

_biomimetics, 2025, doi:10.3390/biomimetics10080506_

Round 1
Reviewer 1 Report
Comments and Suggestions for Authors
The authors present a framework for active EEG signal recording for BCI. In such approach, systems relies on user intentional control, where individuals deliberately focus their attention or perform specific mental tasks to generate distinct brain activity patterns. This approach is a well established method for brainwaves harvesting and classification. It is also considered the easiest approach for data validation.
These patterns classification are detected and translated into commands or actions, if the model is intended to control systems. Active BCIs utilize task-related brain activity, including event-related potentials like imagining hand movements. By engaging users in this process, their brain activity is directly tied to the task or action being performed, allowing for precise control and interaction with the BCI system.
While data harvesting for brainwave classification is one of the many steps towards BCI validation, this step alone doesn't assure smooth translation into real time controlling of systems or devices. Validation and deployment are two very different things within the BCI technology research and development. Validation doesn't necessarily result in successful real time deployment.
It is a good idea for the authors to describe the BCI active vs passive EEG sinal recording. It would be worth for the authors to also add information differentiating validation and deployment, and clarify that their model does the validation of the brainwaves for classification and pattern identification. Otherwise, if real time deployment is done, then the authors should add a session with clear evidence that the model was also deployed in real time.
The recommended points to be addressed by the authors is meant to increase the visibility of the article providing more information that will be beneficial to the readers.
Author Response
#Reviewer 1.
Comments:
The authors present a framework for active EEG signal recording for BCI. In such approach, systems relies on user intentional control, where individuals deliberately focus their attention or perform specific mental tasks to generate distinct brain activity patterns. This approach is a well established method for brainwaves harvesting and classification. It is also considered the easiest approach for data validation.
Reply: While thanking the esteemed reviewer for a thorough review of the manuscript version. We, the authors of the article, believe that your suggestions have been very useful and effective in improving the scientific version of the manuscript. We carefully answered all the questions and suggestions of the esteemed reviewer and added them to the manuscript version.
- These patterns classification are detected and translated into commands or actions, if the model is intended to control systems. Active BCIs utilize task-related brain activity, including event-related potentials like imagining hand movements. By engaging users in this process, their brain activity is directly tied to the task or action being performed, allowing for precise control and interaction with the BCI system.
Reply: Yes, the reviewer's opinion is absolutely correct. This study provides a complete framework for detecting movement intention using EEG signals. Accordingly, a database of movement intentions of the right and left fingers and the resting state has been collected based on EEG signal data. Then, using deep learning methods, these signals are classified into two-class and three-class categories.
- While data harvesting for brainwave classification is one of the many steps towards BCI validation, this step alone doesn't assure smooth translation into real time controlling of systems or devices. Validation and deployment are two very different things within the BCI technology research and development. Validation doesn't necessarily result in successful real time deployment.
Reply: Yes, the respected referee's opinion is absolutely correct. That is why there are various methods for validating educational data. Among these validation methods, we can mention random validation and cross-validation. In this study, following previous studies, random validation was used in evaluating the data.
- It is a good idea for the authors to describe the BCI active vs passive EEG sinal recording. It would be worth for the authors to also add information differentiating validation and deployment, and clarify that their model does the validation of the brainwaves for classification and pattern identification. Otherwise, if real time deployment is done, then the authors should add a session with clear evidence that the model was also deployed in real time.
Reply: Yes, we have two types of processing in recording EEG signals. Processing during online signal recording (real time) and offline signal recording (non-online). This study, following previous studies, uses signal processing in offline mode. In offline processing, the database is first collected. Then, processing and classification will be performed on the signals.
- The recommended points to be addressed by the authors is meant to increase the visibility of the article providing more information that will be beneficial to the readers.
Reply: Based on the reviewer's comments, we have added the offline processing type to the manuscript. We thank the reviewer for his careful comments and the time he has devoted to reviewing the manuscript.

Reviewer 2 Report
Comments and Suggestions for Authors
Dear authors,
Thank you for your research paper.
The paper proposes a hybrid deep learning model combining Graph Convolutional Networks (GCNs) with CNNs for classifying movement intentions from EEG signals in both two- and three-class scenarios. The approach demonstrates notable performance, with classification accuracies of 98% (binary) and 92% (3-classes), outperforming several previous methods. The scientific novelty lies in the integration of graph theory with deep CNNs for EEG signal classification, and the use of a customized architecture optimized via Chebyshev polynomial filters. Additionally, the study emphasizes robustness in noisy environments, which is critical for real-world BCI applications. The evaluation metrics such as accuracy, sensitivity and specificity exceeded 97% in the two-class scenario, and ROC analysis confirmed high discriminatory power. However, while the results obtained are promising, further validation with larger, more diverse datasets and comparative analysis with modern transformer-based architectures would help to confirm the generalizability and comparative power of the proposed method.
While GCNs and CNNs are not new in EEG research, their combination in this specific configuration with effective augmentation via GANs adds value to the research.
However, I have the following observations and recommendations for improving the study:
- The research could benefit from a more in-depth comparison with the most advanced architectures, such as attention-based models or transformers applied to EEG. Therefore, I would recommend comparing your CNN-GCN model with at least one transformer-based model, such as the one described in Xu, Y., Zhang, J., & Liu, X. (2021). EEG-based Emotion Recognition Using Transformer or Mao, X., & Zhong, S. (2022). Temporal-Spatial Transformer for EEG-based Emotion.
- Some of the references, especially in the latter part of the list (e.g. those related to lie detection and general applications of GANs or CNNs), seem less directly related to the topic of the paper. Removing or replacing these references with more directly related EEG BCI references, especially recent ones applying deep learning to motion imagery, would improve the coherence and focus of the reference list.
- Typo on line 15 “ the suggested deep model ”, there is no term "deep model".
- Abstract section. Lines 18-20 Author‘s text “achieving accuracies of 98% in a two-class scenario (left finger taping and right finger taping) and 92% in a three-class scenario (left finger tapping and right finger tapping)”. According to your text it is the same, there is no 3d class.
- Section 2 Materials and procedures. Line 188 Author‘s text “This section begins with a description of the database for ischemic stroke maps”. Why do you mention the ischemic stroke map database? Moreover, the chapter starts with a description of GANs, but not of ischemic stroke maps.
- Line 174 C. Presenting a deep architecture. Here is lacking something, there is no term “deep architecture”
- In Section 2, “Materials and Procedures,” there is only the theoretical foundations of GAN and GNN, but no procedures at all… This looks like a discrepancy between what was planned and what was done. This needs to be explained, otherwise it looks like a logical inconsistency.
- Section 3 Proposed model. Line 286, What is meant by “database recording”, I assume it is a data collection?
- Subsection 3.2. Pre-processing of EEG data.Lines 328, 329 “The data pertaining to channels F3-C3, Fz-Cz, F4-C4, C3-P3, Cz-Pz, and C4-328 P4 were extracted from the 50 Hz frequency of municipal electricity utilizing a notch filter.” I would recommend reframing this sentence so that the 50 Hz artifact was removed using a notch filter.
- Line 458: It is unclear whether you mean your research with the phrase “This research utilizes pre-trained networks Inception [36], VGG [37], U-net [38] and basic CNN [28]. “If so, you have not described the models used.
- Subsection 4.3. Comparison with current methodologies and research, line 494, Table 4. In Table 4, you described that you used GAN+graph theory and CNN, but in Section 2, “Materials and Procedures,” you only described the GAN and GNN models in detail, and there is still no explanation of how exactly you came up with the combination of “GAN+graph theory and CNN.”
Best regards,
reviewer
Comments on the Quality of English Language
The English is generally understandable but suffers from noticeable grammatical and stylistic issues throughout. Examples include repeated phrases "automatic deep model for automatic detection", awkward constructions, inconsistent terminology ("finger taping" vs. "finger tapping"), and overuse of passive voice. Technical clarity can also be improved with better sentence structuring and removal of redundancies. Overall, the language is serviceable but would benefit greatly from thorough editing by a native English speaker or a professional editor.
Author Response
#Reviewer 2.
Comments:
The paper proposes a hybrid deep learning model combining Graph Convolutional Networks (GCNs) with CNNs for classifying movement intentions from EEG signals in both two- and three-class scenarios. The approach demonstrates notable performance, with classification accuracies of 98% (binary) and 92% (3-classes), outperforming several previous methods. The scientific novelty lies in the integration of graph theory with deep CNNs for EEG signal classification, and the use of a customized architecture optimized via Chebyshev polynomial filters. Additionally, the study emphasizes robustness in noisy environments, which is critical for real-world BCI applications. The evaluation metrics such as accuracy, sensitivity and specificity exceeded 97% in the two-class scenario, and ROC analysis confirmed high discriminatory power. However, while the results obtained are promising, further validation with larger, more diverse datasets and comparative analysis with modern transformer-based architectures would help to confirm the generalizability and comparative power of the proposed method.
While GCNs and CNNs are not new in EEG research, their combination in this specific configuration with effective augmentation via GANs adds value to the research.
Reply: While thanking the esteemed reviewer for a thorough review of the manuscript version. We, the authors of the article, believe that your suggestions have been very useful and effective in improving the scientific version of the manuscript. We carefully answered all the questions and suggestions of the esteemed reviewer and added them to the manuscript version.
However, I have the following observations and recommendations for improving the study:
- The research could benefit from a more in-depth comparison with the most advanced architectures, such as attention-based models or transformers applied to EEG. Therefore, I would recommend comparing your CNN-GCN model with at least one transformer-based model, such as the one described in Xu, Y., Zhang, J., & Liu, X. (2021). EEG-based Emotion Recognition Using Transformer or Mao, X., & Zhong, S. (2022). Temporal-Spatial Transformer for EEG-based Emotion.
Reply: Yes, the respected referee's opinion is absolutely correct, in this regard, we have added study Xu, Y., Zhang, J., & Liu, X. (2021). EEG-based Emotion Recognition Using Transformer, which is related to emotion recognition from brain signals, to the manuscript. However, we did not find your second study for review. Please send us its DOE link so that we can review that study in the manuscript as well.
Which are highlighted on page 1 of the introduction section, Ref [2].
- Some of the references, especially in the latter part of the list (e.g. those related to lie detection and general applications of GANs or CNNs), seem less directly related to the topic of the paper. Removing or replacing these references with more directly related EEG BCI references, especially recent ones applying deep learning to motion imagery, would improve the coherence and focus of the reference list.
Reply: Yes, the respected referee's opinion is correct. In this regard, we have added references to the years 2024 and 2025 to the manuscript.
- Typo on line 15 “ the suggested deep model ”, there is no term "deep model".
Abstract section. Lines 18-20 Author‘s text “achieving accuracies of 98% in a two-class scenario (left finger taping and right finger taping) and 92% in a three-class scenario (left finger tapping and right finger tapping)”. According to your text it is the same, there is no 3d class.
Reply: Yes, the respected referee's opinion is absolutely correct. In accordance with the respected referee's opinion, the abstract text has been rewritten as follows:
“Following the requisite pre-processing, the captured signals are input into the proposed model, which is constructed on the integration of graph theory and deep convolutional networks.”
and
“In this study, we introduce a novel architecture based on six deep convolutional graph layers, specifically designed to effectively capture and extract essential features from EEG signals. The proposed model demonstrates remarkable performance, achieving an accuracy of 98% in a bina-ry classification task distinguishing between left and right finger tapping. Furthermore, in a more complex three-class classification scenario, which includes left finger tapping, right finger tap-ping, and an additional class, the model attains an accuracy of 92%. These results highlight the effectiveness of the architecture in decoding motor-related brain activity from EEG data.”
Which are highlighted on page 1 of the abstract section, line 15-23.
- Section 2 Materials and procedures. Line 188 Author‘s text “This section begins with a description of the database for ischemic stroke maps”. Why do you mention the ischemic stroke map database? Moreover, the chapter starts with a description of GANs, but not of ischemic stroke maps.
Reply: Yes, the respected referee's opinion is absolutely correct. There was an inadvertent typo in this section, which, with the careful attention of the respected referee, has been changed to the following text:
“In this section, the mathematical basis of the algorithms used in this research, which include Generative adversarial networks (GAN) and graph neural networks (GNN), is fully examined.”
Which are highlighted on page 1 of the abstract section, line 15-23.
- Line 174 C. Presenting a deep architecture. Here is lacking something, there is no term “deep architecture”
Reply: Yes, the respected referee's opinion is absolutely correct. Accordingly, we have rewritten the relevant sentence as follows:
“Presenting a new model with high speed and accuracy for classifying left finger stroke, right finger stroke and resting state.”
Which are highlighted on page 1 of the abstract section, line 15-23.
- In Section 2, “Materials and Procedures,” there is only the theoretical foundations of GAN and GNN, but no procedures at all… This looks like a discrepancy between what was planned and what was done. This needs to be explained, otherwise it looks like a logical inconsistency.
Reply: Y With respect to the opinion of the esteemed referee, in this research, we have only used generative adversarial network and convolutional graph network algorithms for 2-class and 3-class movement intention classification. Accordingly, we have presented the mathematical foundations of these algorithms in Section 2.
- Section 3 Proposed model. Line 286, What is meant by “database recording”, I assume it is a data collection?
Reply: Yes, it means data recording. Accordingly, we changed the relevant sentence to data recording.
- Subsection 3.2. Pre-processing of EEG data.Lines 328, 329 “The data pertaining to channels F3-C3, Fz-Cz, F4-C4, C3-P3, Cz-Pz, and C4-328 P4 were extracted from the 50 Hz frequency of municipal electricity utilizing a notch filter.” I would recommend reframing this sentence so that the 50 Hz artifact was removed using a notch filter.
Reply: Based on the opinion of the esteemed referee, the relevant sentence has been rewritten as follows:
“To eliminate the interference caused by the 50 Hz frequency of municipal electricity, a notch filter was applied to the EEG data collected from the F3-C3, Fz-Cz, F4-C4, C3-P3, Cz-Pz, and C4-P4 channel pairs.”
- Line 458: It is unclear whether you mean your research with the phrase “This research utilizes pre-trained networks Inception [36], VGG [37], U-net [38] and basic CNN [28]. “If so, you have not described the models used.
Reply: With respect to the opinion of the esteemed referee, pre-trained models including VGG, u-net, CNN and etc. are well-known models that are widely compared in various studies. Accordingly, a separate explanation of these models would unnecessarily increase the number of pages of the article and would tire the readers. However, if the referee feels the need to explain these models, these models can be explained in Section 2.
- Subsection 4.3. Comparison with current methodologies and research, line 494, Table 4. In Table 4, you described that you used GAN+graph theory and CNN, but in Section 2, “Materials and Procedures,” you only described the GAN and GNN models in detail, and there is still no explanation of how exactly you came up with the combination of “GAN+graph theory and CNN.”
Reply: With respect to the reviewer's opinion, our model is a combination of adversarial generative networks, graph theory, and CNN networks. This model has been developed through trial and error and has been compared with recent popular networks and has achieved promising results. In Section 2, where the GNN model is discussed, it is the same combination of graph theory and CNN networks that is meant.
Best Regards